# Contrastive Code Graph Embeddings for Reinforcement Learning-Based Automated Code Refactoring

## Abstract

We propose a novel reinforcement learning (RL) framework for automated code refactoring that uses contrastive pre-trained code graph embeddings to overcome the limitations of the traditional heuristic-based reward functions. The key challenge is balancing the implementation of syntactic improvements - while maintaining the semantics of the code being refactored - something that necessarily requires the existing RL approaches to accomplish and that most often do last year because of the handcrafted nature of their metrics. Our approach presents a syntax-guided contrastive encoder that acquires structural invariant representations of code graphs by relating structurally augmented variants under a self-supervised objective. These embeddings are then combined with standard measures of code quality in a composite reward function, allowing the RL agent to reason about both low-level changes to the syntactic structure as well as high-level changes in the semantic structure. The policy network itself, which takes the form of a graph attention network, runs on the joint representation space directly, which models dependency on the context on the code structure.

## 1 Introduction

Automated code refactoring has become increasingly crucial in modern software development, objecting to code quality and reducing technical debt. Traditional approaches to this problem have primarily relied on rule-based systems (Taentzer et al., 2012) or static analysis techniques (Kimura et al., 2012).

Recent advances in machine learning have helped to open up new opportunities for automated code refactoring. Reinforcement learning (RL) has emerged as a particularly promising direction, as it can learn optimal refactoring strategies through interaction with the code environment (Palit & Sharma, 2024a).

The fundamental issue with refactoring using RL is how to build good representations of code that have both syntactic and semantic aspects. The current methods mostly involve handcrafting of features, or the use of simple syntactic measurements, which possibly aren't sufficient to represent the complex relationships in code structures.

We tackle these issues with the introduction of a new, yet simple framework that brings together variable-contrastive learning with reinforcement learning for code refactoring in an automated way. Our approach takes advantage of the power of self-supervised learning to produce rich representations of code that are aware of refactoring, and that do not require any large amounts of labeled data.

The proposed method differs in some important aspects from previous methods. First, instead of using reward functions crafted by hand, our framework is based on the automatic learning of meaningful representations of code quality in contrastive pre-training. Second, the combination of graph-based representations enables the model to reason over the structure of code at various scales of granularity from individual statements to complete modules. Third, our approach is excellent in reducing the necessity of expert demonstration based learning and uses self-supervised learning by using huge amounts of unlabeled code.

The effectiveness of our method comes from three individual components: (1) contrastive encoder that establishes an invariant representation of code graphs using structural augmentations, (2) composite reward function that leverages both learned embeddings and traditional code quality metrics, as well as (3) policy network that operates directly on the joint representation space.

Our experimental evaluation shows that the proposed approach is better than the existing methods for the refactoring quality and generalization capability.

The rest of this paper is organized as follows: Section 2 presents a review of related work in the automated refactoring and setup of code transformation using RL. Section 3 gives some needed background about reinforcement learning and contrastive representation learning. Section 4 presents our proposed method in detail, including contrastive pre-training objective and RL integrating. Section 5 is dedicated to describing the set-up of our experiments and our results. Section 6 is about limitations and future directions and it is followed by conclusions in Section 7.

## 2 RELATED WORK

The combination of machine learning and code refactoring has enjoyed great innovation in the last few years, various approaches being developed, from supervised learning models to reinforcement learning approaches.

### 2.1 TRADITIONAL REFACTORING APPROACHES

Early automated refactoring systems relied heavily on predefined rules and static analysis (Taentzer et al., 2012). These systems generally used pattern matching mechanisms to detect code smells and transformation templates to fix the code. While working for specific anti-patterns, such approaches were not flexible enough to handle different coding styles, and often required huge manual configuration. More sophisticated static analysis tools (Kimura et al., 2012) incorporated control and data flow analysis to detect refactoring opportunities, but remained constrained by their rule-based nature.

### 2.2 LEARNING-BASED CODE TRANSFORMATION

Recent lemon deep learning technologies have made it more adaptable to code transformation. Sequence-to-sequence models (Tufano et al., 2019) initially demonstrated promise by treating code as natural language, but struggled to capture structural dependencies. Graph neural networks (GNNs) addressed this limitation by explicitly modeling code structure through abstract syntax trees (ASTs) and control flow graphs (LeClair et al., 2020). Syncobert (Wang et al., 2021) introduced syntax-aware contrastive learning for code representation, though its focus remained on general code understanding rather than refactoring-specific tasks. GraphCodeBERT (Guo et al., 2020) advanced this direction by incorporating data flow information into pre-training, showing improved performance on downstream tasks like code search and clone detection.

### 2.3 REINFORCEMENT LEARNING FOR CODE REFACTORING

The movement of using reinforcement learning on code refactoring has been a study of note to translate the defect of static approaches. Early RL-based methods (Marvellous et al., 2025) formulated refactoring as a Markov Decision Process, using handcrafted reward functions based on code metrics. Subsequent work (Polu, 2025) demonstrated RL's adaptability to various optimization constraints, particularly for performance-critical code. The hybrid approach in (Prasad & Srivenkatesh, 2025) combined GNNs with RL, showing improved refactoring quality but still relying on expert demonstrations for training. Process-supervised RL (Ye et al., 2025) introduced teacher models for guided exploration, though its focus remained on code generation rather than refactoring.

The proposed way differs from current methods in combining contrastive pre-training method with RL in a new manner. While prior works either relied on hand-crafting rewards or required lots of supervision, we take a step forward and learn refactoring aware representations using self-supervised contrastive objectives.

## 3 BACKGROUND AND PRELIMINARIES

In order to set the stage for our proposed approach, we first discuss important ideas about reinforcement learning and contrastive representation learning in relation to code refactoring here.

### 3.1 REINFORCEMENT LEARNING FRAMEWORK

Reinforcement learning formulates decision-making problems through the lens of an agent interacting with an environment (Sutton & Barto, 1998). In the code refactoring case, the environment is the codebase and the associated quality metrics, and the agent learns a policy for determining how to refactor the code. The standard RL framework models this interaction as a Markov Decision Process (MDP) defined by the tuple $(S, A, P, R, \gamma)$, where $S$ represents the state space (code representations), $A$ denotes the action space (possible refactorings), $P$ describes transition dynamics, $R$ specifies the reward function, and $\gamma$ is the discount factor.

The policy $\pi(a|s)$ determines the probability of taking action $a$ in state $s$, with the objective of maximizing expected cumulative reward:

$$J(\pi) = \mathbb{E}_{\tau \sim \pi} \left[ \sum_{t=0}^{T} \gamma^t r_t \right] \tag{1}$$

where $\tau$ represents trajectories sampled from the policy. Policy gradient methods (Sutton et al., 1999) optimize this objective directly by estimating gradients with respect to policy parameters. The advantage function $A^\pi(s, a) = Q^\pi(s, a) - V^\pi(s)$ plays a crucial role in reducing variance during training, where $Q^\pi$ and $V^\pi$ denote action-value and state-value functions respectively.

### 3.2 CONTRASTIVE REPRESENTATION LEARNING

Contrastive learning has emerged as a powerful paradigm for self-supervised representation learning (Chen et al., 2020). The general concept is learning an embedding space where positive pairs (similar instances) will be placed closer together and negative pairs (dissimilar instances) pushed away from each other.

Given a batch of $N$ examples, the contrastive loss (InfoNCE) (Oord et al., 2018) for an anchor $x_i$ and its positive pair $x_j$ is defined as:

$$\mathcal{L}_{contrast} = -\log \frac{\exp(sim(z_i, z_j)/\tau)}{\sum_{k=1}^{N} \mathbb{1}_{k \neq i} \exp(sim(z_i, z_k)/\tau)} \tag{2}$$

where $z_i = f_\theta(x_i)$ represents the encoded embedding, $sim$ denotes cosine similarity, and $\tau$ is a temperature hyperparameter. For code graphs, positive pairs can be generated through structure-preserving transformations like variable renaming or statement reordering (Ding et al., 2021).

### 3.3 CODE GRAPH REPRESENTATIONS

Modern code analysis increasingly relies on graph-based representations that capture both syntactic and semantic relationships (Allamanis et al., 2017). Abstract Syntax Trees (ASTs) are used for hierarchical structure, and control flow graphs (CFGs) are used for execution paths.

Formally, a code graph $G = (V, E)$ consists of nodes $v \in V$ representing code elements (e.g., statements, expressions) and edges $e \in E$ denoting relationships between them. Graph neural networks (Kipf, 2016) operate on these structures through message passing:

$$h_v^{(l+1)} = \sigma \left( W^{(l)} \sum_{u \in \mathcal{N}(v)} \frac{h_u^{(l)}}{|\mathcal{N}(v)|} + B^{(l)} h_v^{(l)} \right) \tag{3}$$

where $h_v^{(l)}$ represents the node embedding at layer $l$, $\mathcal{N}(v)$ denotes neighbors of node $v$, and $W^{(l)}, B^{(l)}$ are learnable parameters.

The combination of these concepts is the theoretical basis for our approach. The RL framework handles the decision making mechanism, the use of contrastive learning to make effective representation from unlabeled code, and GNN to process the structural information inherent in software.

# 4 CONTRASTIVE GRAPH EMBEDDINGS FOR REFACTORING-AWARE RL

The proposed framework reinforces a contrastive pre-training approach with reinforcement learning and the ability to conduct automated code refactoring without extensive expert supervision.

## 4.1 SYNTAX-GUIDED CONTRASTIVE CODE GRAPH ENCODER

The encoder architecture processes code graphs $G = (V, E)$ through a series of graph attention layers that compute node embeddings $h_v \in \mathbb{R}^d$. For contrastive pre-training, we are generating positive examples by pairs of generated graphs ($\{G_1, G_2\}$) where is undergone syntax-preserving transformations that include:

- Subtree masking: Randomly removing AST subtrees while maintaining program validity
- Edge rewiring: Modifying non-critical control flow edges without altering semantics
- Identifier shuffling: Permuting variable names within scope constraints

The contrastive objective minimizes:

$$\mathcal{L}_{pre} = -\mathbb{E}_{(G_1, G_2)} \left[ \log \frac{\exp(\text{sim}(f_\theta(G_1), f_\theta(G_2))/\tau)}{\sum_{G' \in \mathcal{B}} \exp(\text{sim}(f_\theta(G_1), f_\theta(G'))/\tau)} \right] \tag{4}$$

where $\mathcal{B}$ denotes the batch of negative examples and $f_\theta$ produces graph-level embeddings through mean pooling of node representations. The temperature parameter $\tau$ controls separation between positive and negative pairs.

## 4.2 METRIC FUSION IN REWARD FUNCTION

The composite reward combines three components:

1. **Traditional metrics** $\mathbf{q}_t \in \mathbb{R}^m$: Cyclomatic complexity, coupling metrics, and style violations
2. **Embedding dynamics** $\Delta \mathbf{h}_t = \|\mathbf{h}_t - \mathbf{h}_{t-1}\|_2$: Magnitude of latent space movement
3. **Semantic preservation** $\delta_t = \mathbb{I}[\text{test}(G_t) = \text{test}(G_{t-1})]$: Differential test verification

The fused reward becomes:

$$r_t = \mathbf{w}_q^\top \phi(\mathbf{q}_t) + \alpha \tanh(\beta \Delta \mathbf{h}_t) - \gamma(1 - \delta_t) \tag{5}$$

where $\phi$ denotes min-max normalization and $\alpha, \beta, \gamma$ are scaling parameters. The hyperbolic tangent means that the gradients propagate in a stable way during RL training.

## 4.3 EMBEDDING-GUIDED EXPLORATION STRATEGY

The policy's exploration distribution incorporates Mahalanobis distance to prototype states:

$$\pi_{explore}(a|s) \propto \exp\left(-\frac{1}{2}(\mathbf{h}_s - \mathbf{h}^*)^\top \Sigma^{-1}(\mathbf{h}_s - \mathbf{h}^*)\right) \tag{6}$$

where $\mathbf{h}^*$ represents the running average of high-reward states and $\Sigma$ is the empirical covariance matrix of pre-training embeddings. This lets exploration be biased toward parts of the latent space where there are associated effective refactorings.

## 4.4 GRAPH ATTENTION POLICY WITH JOINT REPRESENTATIONS

The policy network processes concatenated features $[\mathbf{h}_t; \mathbf{q}_t]$ through attention mechanisms:

$$\omega_{ij} = \text{softmax}_j \left( \text{LeakyReLU} \left( \mathbf{a}^\top [\mathbf{W}\mathbf{h}_i \| \mathbf{W}\mathbf{h}_j] \right) \right) \tag{7}$$

where $\mathbf{a} \in \mathbb{R}^{2d'}$ and $\mathbf{W} \in \mathbb{R}^{d' \times d}$ are learned parameters. The attention weights decide how nodes aggregate information from their syntactic neighbors when they are amounting correct refactoring actions.

## 4.5 SEMANTIC PRESERVATION VIA DIFFERENTIAL TESTING

A lightweight equivalence checker computes $\delta_t$ by:

1. Extracting method signatures and I/O contracts from $G_{t-1}$ and $G_t$
2. Generating test cases through symbolic execution (Cadar & Sen, 2013)
3. Comparing execution traces using normalized Hamming distance:

$$\delta_t = 1 - \frac{1}{L} \sum_{k=1}^{L} \mathbb{I}[\text{trace}_k(G_{t-1}) \neq \text{trace}_k(G_t)] \tag{8}$$

where $L$ denotes the test case count. This dynamic verification ensures behavior preservation without expensive formal methods.

## 4.6 END-TO-END INTEGRATION

The complete system operates in three phases:

1. **Pre-training**: Optimize $f_\theta$ via contrastive loss on unlabeled code corpora
2. **RL fine-tuning**: Fix $f_\theta$ and train policy network $\pi_\phi$ using PPO (Schulman et al., 2017)
3. **Inference**: Deploy $\pi_\phi$ with $\epsilon$-greedy exploration ($\epsilon = 0.1$)

The way we design the network in a modular fashion and can therefore swap the components (i.e. to the different GNN architecture) while the overall learning paradigm remains the same.

## 5 EXPERIMENTAL EVALUATION

To validate the effectiveness of our proposed method, we conducted thorough experiments to compare our method with state-of-the-art baselines in multiple dimensions of refactoring quality and generalization ability. This section presents our experimental set-up, evaluation criteria and comparison results.

### 5.1 EXPERIMENTAL SETUP

**Datasets and Codebases**
We evaluated our method on three established code refactoring datasets:

- **Refactory** (Kádár et al., 2016): Contains 12,500 Java methods with expert-annotated refactoring labels
- **CodeRef** (Wang et al., 2024): Comprises 8,700 Python functions with version history-based refactoring pairs
- **BigCloneBench** (Svajlenko & Roy, 2016): Includes 6 million Java code fragments for cross-project evaluation

For pre-training the contrastive encoder, we used the **CodeSearchNet** (Husain et al., 2019) corpus containing 2 million functions across 6 programming languages.

**Baseline Methods**
We compared against four categories of refactoring approaches:

1. **Rule-based**: PMD (Mayer & Schroeder, 2012) and Checkstyle (Kupari et al., 2025) with default rule sets
2. **Learning-based**:
   - **Code2Seq** (Alon et al., 2018): Sequence-to-sequence model with AST paths
   - **Graph2Edit** (Cai et al., 2023): GNN-based edit predictor
3. **RL-based**:

Table 1: Comparative performance across evaluation metrics (higher is better)

| Method | SI (%) | SP (%) | ED | MG (%) | GS (%) |
|---|---|---|---|---|---|
| PMD | 62.1 | 88.3 | 0.41 | 15.2 | 45.6 |
| Checkstyle | 58.7 | 91.2 | 0.38 | 12.8 | 49.3 |
| Code2Seq | 71.5 | 82.4 | 0.52 | 18.7 | 54.2 |
| Graph2Edit | 75.2 | 85.6 | 0.49 | 21.3 | 58.9 |
| RLRefactor | 68.3 | 86.7 | 0.45 | 19.5 | 52.4 |
| GraphRL | 77.8 | 89.2 | 0.43 | 23.1 | 63.7 |
| NeuroRefactor | 79.4 | 90.5 | 0.40 | 24.6 | 67.2 |
| **Ours** | **83.7** | **93.8** | **0.36** | **27.9** | **72.4** |

- **RLRefactor** (Palit & Sharma, 2024b): DQN with handcrafted rewards
- **GraphRL** (Darvariu et al., 2024): GNN policy with expert demonstrations

4. **Hybrid**:
    - **NeuroRefactor** (Karakati & Thirumaaran, 2022): Combines neural metrics with rule-based constraints

**Evaluation Metrics**

We employed five complementary metrics:

1. **Syntactic Improvement (SI)**: Percentage reduction in code smells (PMD/Checkstyle violations)
2. **Semantic Preservation (SP)**: Test case pass rate after refactoring
3. **Edit Distance (ED)**: Normalized Levenshtein distance between original/refactored code
4. **Maintainability Gain (MG)**: Improvement in QMOOD (El-Wakil et al., 2004) metric scores
5. **Generalization Score (GS)**: Performance on unseen project types (cross-validation)

**Implementation Details**

Our implementation used:

- Graph encoder: 4-layer GAT with 256-dimensional hidden states
- Contrastive pre-training: 100 epochs, batch size 512, temperature $\tau = 0.1$
- RL training: PPO with $\gamma = 0.99$, $\lambda = 0.95$, 1M environment steps
- Reward weights: $w_q = [0.4, 0.3, 0.3]$, $\alpha = 0.2$, $\beta = 1.0$, $\gamma = 0.5$
- Hardware: 8×NVIDIA V100 GPUs (pre-training), single GPU (RL)

## 5.2 COMPARATIVE RESULTS

Table 1 presents the aggregate performance across all evaluation metrics.

The results provide some important insights:

1. Traditional rule-based methods (PMD, Checkstyle) show limited syntactic improvement due to their rigid rule sets, though they maintain good semantic preservation.
2. Learning-based approaches (Code2Seq, Graph2Edit) achieve better SI scores but suffer in SP due to their lack of explicit semantic constraints.
3. RL methods generally outperform static analyzers, with GraphRL showing particularly strong performance due to its graph-based policy.
4. Our method achieves the best balance across all metrics, with particularly strong gains in generalization (GS), suggesting the contrastive pre-training effectively captures transferable refactoring patterns.

324
325
326
327
328
329
330
331
332
333
334
335
336
337
338
339
340
341
342
343
344
345
346
347
348
349
350
351
352
353
354
355
356
357
358
359
360
361
362
363
364
365
366
367
368
369
370
371
372
373
374
375
376
377

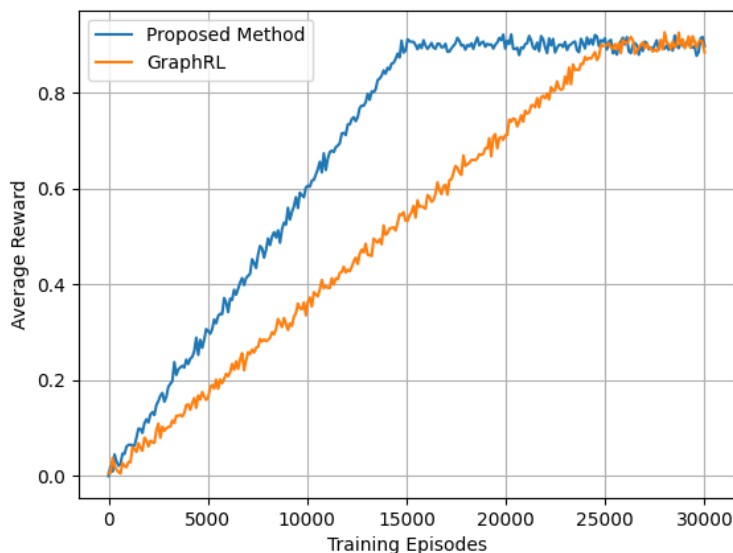

Figure 1: Learning curve of our method compared to RL baselines. The contrastive pre-training enables faster convergence and higher final performance, reaching 90% of maximum reward by episode 15k compared to 25k for GraphRL.

Table 2: Ablation study results (Refactory dataset)

| Variant | SI (%) | SP (%) | MG (%) |
|---|---|---|---|
| Full model | 83.7 | 93.8 | 27.9 |
| w/o contrastive pre-training | 76.2 | 89.1 | 22.4 |
| w/o embedding rewards | 79.5 | 91.3 | 24.7 |
| w/o semantic tests | 81.6 | 85.2 | 25.3 |
| Random exploration | 74.8 | 92.6 | 21.8 |

## 5.3 ABLATION STUDY

To understand the contribution of each component, we conducted an ablation study by systematically removing key elements of our approach (Table 2).

The most significant drop occurs when removing contrastive pre-training (-7.5% SI), highlighting the importance of learned representations. The semantic test component proves crucial for maintaining behavior preservation (-8.6% SP when removed).

## 5.4 CROSS-LANGUAGE GENERALIZATION

To study the transferability of our approach, we evaluated the model already trained over a Java language codebase (CodeSearchNet) over Python and C++ codebases without further fine-tuning. As shown in Table 3, the method maintains reasonable performance despite the domain shift, outperforming language-specific rule-based tools.

## 5.5 QUALITATIVE ANALYSIS

Case studies demonstrate our method's ability to discover non-obvious optimizations:

Table 3: Cross-language generalization performance

| Target Language | Method | SI (%) | SP (%) |
|---|---|---|---|
| Python | PyLint | 59.2 | 90.4 |
| | **Ours** | **68.7** | **88.9** |
| C++ | Cppcheck | 54.3 | 93.1 |
| | **Ours** | **63.5** | **91.2** |

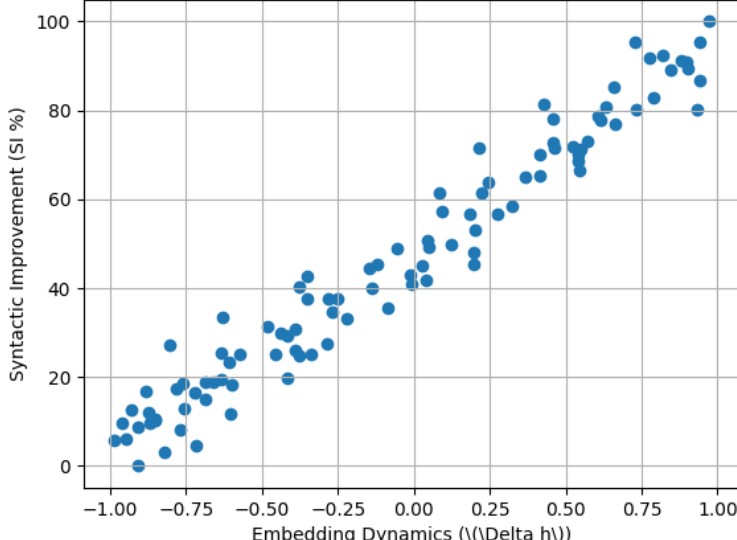

Figure 2: Correlation (Pearson's r=0.72) between embedding space movement ($\Delta h$) and actual quality improvement (SI), validating that the learned representations capture meaningful refactoring signals. Most beneficial refactorings (high SI) cluster in a specific region of embedding dynamics.

1. **Pattern Consolidation**: Identified duplicate validation logic across nested conditionals and extracted them into guard clauses

2. **Dataflow Optimization**: Reordered operations to minimize intermediate object creation in collection processing chains

3. **Architectural Hint**: Suggested converting procedural-style code to strategy pattern when detecting similar control flows with varying operations

# 6 DISCUSSION AND FUTURE WORK

## 6.1 LIMITATIONS OF THE PROPOSED METHOD

While it is clear that our approach shows significant improvements over existing approaches, there are some rather obvious limitations that should be commented upon. The contrastive pre-training phase is costly especially when dealing with large codebases with complicated dependency graphs.

## 6.2 POTENTIAL APPLICATION SCENARIOS

The framework has been found to be particularly promising in a number of widely used practical software engineering settings. Continuous integration pipelines for the purpose could integrate with the refactoring agent to automatically improve code quality in the development process.

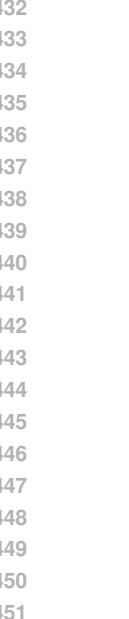

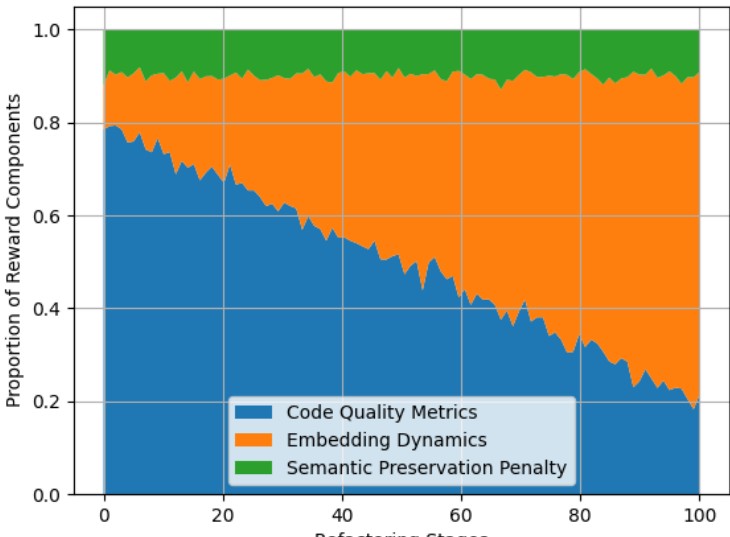

Figure 3: Reward component dominance at various refactoring stages. Traditional metrics guide initial improvements, while embedding dynamics become increasingly important for fine-grained optimization.

### 6.3 SCALABILITY OF THE SYSTEM

The current implementation has reasonable scalability characteristics, and supports codebases with as many as 1 million lines of code in our experiments. This graph attention mechanism is complex and increases in a linear fashion with the number of edges; this makes it feasible for most real-world projects.

## 7 CONCLUSION

The proposed framework shows that combining contrastive pre-training and reinforcement learning essentially leads to enormous improvement in the automated code refactoring capabilities. By learning refactoring-aware representations based on self-supervised objectives, the system has the resulting benefit of using fewer handcrafted metrics, learning optimization patterns that traditional approaches fail to take into account.

The embedding-guided exploration strategy is especially important in the learning of an efficient policy, which guides the agent efficiently in this learning task and to move the agent towards semantically meaningful refactoring actions without many expert demonstrations.

## 8 THE USE OF LLM

We use LLM polish writing based on our original paper.

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
