# OpenReview forum: "Contrastive Code Graph Embeddings for Reinforcement Learning-Based Automated Code Refactoring"
_ICLR.cc/2026/Conference — Submitted to ICLR 2026_

### Official Review · Reviewer_BKDb · 2025-10-30

**Soundness:** 1
**Presentation:** 1
**Contribution:** 1
**Rating:** 0
**Confidence:** 4

**Summary:**

The manuscript strongly resembles AI-generated content and may have been produced as an internal test for prospective AI researchers. If so, it suggests that the current state of such roles remains immature and requires further development.

**Strengths:**

I believe this paper was generated by AI. If not, please let me know.

**Weaknesses:**

I believe this paper was generated by AI. If not, please let me know.

**Questions:**

See Weaknesses.

---

### Official Review · Reviewer_qRvF · 2025-10-31

**Soundness:** 2
**Presentation:** 2
**Contribution:** 1
**Rating:** 2
**Confidence:** 4

**Summary:**

The paper’s main contribution is introducing a contrastive pre-training and reinforcement learning (RL) framework for automated code refactoring. It proposes a syntax-guided contrastive encoder that learns structural-invariant code graph embeddings using transformations like subtree masking, edge rewiring, and identifier shuffling. These embeddings are combined with conventional code-quality metrics (e.g., cyclomatic complexity, coupling) into a composite reward function, enabling the RL agent to balance syntactic improvements with semantic preservation. A graph attention network (GAT) policy operates on this joint representation space, with an embedding-guided exploration strategy that biases actions toward high-reward latent states. Experiments on multiple code-refactoring benchmarks show improved syntactic improvement, semantic preservation, and generalization compared with rule-based, learning-based, and previous RL baselines.

**Strengths:**

Originality: Creative combination of syntax-guided contrastive pre-training on code graphs with RL for refactoring, including semantics-preserving augmentations and an embedding-guided exploration bias, an interesting twist beyond hand-crafted rewards alone.


Clarity: The paper lays out a modular pipeline and presents core equations for message passing and contrastive loss, which helps readers follow the design even if prose needs polish.

**Weaknesses:**

Contrastive pre-training is under-specified.
The paper names three augmentations and gives an InfoNCE objective, but omits crucial details: augmentation probabilities/ratios, semantic-preservation checks for positives, projection head design, optimizer/LR schedule, and negative sampling strategy. Actionable: add a “Pretraining Details” appendix with full recipe.

No standalone evaluation of the learned embeddings.
Pre-training quality is inferred only indirectly via downstream ablation; there’s no linear probe, retrieval, clustering, or probe tasks on code-understanding benchmarks. Actionable: add pretraining-only evaluations before RL. The current Experiment section list does not include a pretraining evaluation subsection.

Composite reward lacks operational clarity.
Eq. (5) defines a fused reward but does not specify the exact metrics vector, normalization windows, or how weights (w_q, α, β, γ) were chosen/tuned; no sensitivity study is reported. Actionable: precisely enumerate metrics (and implementations), describe normalization (per-project or global), report a small grid/sensitivity study, and include per-component ablations on at least one dataset.


Positioning/novelty vs. close baselines is thin.
The approach is compared to rule-based, learned, and RL baselines, but there’s no GraphCodeBERT + identical RL head baseline to isolate the value of this contrastive pretraining recipe. Actionable: add baselines where you swap in public code graph encoders (e.g., GraphCodeBERT) under the same RL stack, and discuss where your gains come from (augmentations? reward fusion?).

**Questions:**

Contrastive Pre-training Clarity:
Could the authors provide full details of the contrastive pre-training process, specifically the architecture of the projection head, augmentation ratios and constraints, optimizer settings, and how semantic equivalence between augmented graph pairs was validated? The current description (Section 4.1) lists transformations but lacks implementation-level transparency.

Evaluation of Learned Embeddings:
How do you verify that the pre-trained embeddings capture meaningful structural and semantic code relationships before reinforcement learning fine-tuning? No standalone embedding evaluation is presented, making it unclear whether contrastive pre-training itself contributes to downstream performance.

Novelty and Baseline Comparison:
Given that models like GraphCodeBERT already perform syntax-guided contrastive pre-training for code representation, what are the novel aspects of your method beyond integrating the embeddings into the RL reward? Have you compared your approach directly against those encoders under identical RL settings to isolate the impact of your proposed pre-training?

---

### Official Review · Reviewer_4u9X · 2025-11-06

**Soundness:** 2
**Presentation:** 2
**Contribution:** 2
**Rating:** 4
**Confidence:** 3

**Summary:**

This paper presents a framework for automated code refactoring utilizing reinforcement learning (RL). The core contribution is a composite reward function that guides an RL agent, intended to replace traditional handcrafted metrics. This function combines (1) contrastive pre-trained code graph embeddings, (2) standard code quality metrics, and (3) a differential test-based semantic preservation score. The agent's policy is represented by a graph attention network  that operates on code graph representations.

**Strengths:**

The paper addresses an important and practical problem: the automated refactoring of code to improve quality and reduce technical debt. The core motivation to move beyond purely heuristic-based reward functions and incorporate learned, semantically-aware representations  is a valid and promising research direction. The proposed framework, which integrates contrastive pre-training, a graph-based policy network, and explicit semantic checks, is in concept, comprehensive.

**Weaknesses:**

1. Feasibility: The most significant weakness is the methodological soundness of the semantic preservation component. The claim of using symbolic execution —an algorithm known for its high computational complexity—as a "lightweight" reward signal in an RL loop  is extraordinary and requires substantial justification. Without a detailed explanation of how this is achieved tractably, the entire experimental results section  is called into question.
2. Unclear Reward Formulation: The rationale for including "Embedding dynamics" as a positive reward component is not well-justified. The paper does not provide a clear theoretical or empirical argument for why a large magnitude of movement in the latent space should inherently correspond to a high-quality refactoring step. This component feels arbitrary and is insufficiently motivated.
3. Poor Presentation: The text is replete with syntactically awkward constructions, and unclear sentences, which severely impede comprehension. This lack of clarity obscures the paper's technical details and makes a thorough evaluation difficult.

**Questions:**

see weaknesses

---

### Official Review · Reviewer_EV4W · 2025-11-12

**Soundness:** 2
**Presentation:** 2
**Contribution:** 1
**Rating:** 2
**Confidence:** 3

**Summary:**

This paper proposes a framework called Contrastive Code Graph Embeddings (CCGE) for reinforcement learning-based automated code refactoring. The authors aim to overcome the limitations of hand-engineered reward functions by leveraging contrastive pre-training on code graphs and incorporating the learned embeddings into a composite reward function that also accounts for syntactic and semantic correctness. The policy network is implemented as a graph attention network (GAT) operating on the learned joint representation.
Experiments are conducted on several code refactoring datasets (Refactory, CodeRef, BigCloneBench), showing improvements over traditional static, learning-based, and prior RL baselines in terms of syntactic improvement, semantic preservation, and generalization.

**Strengths:**

Timely and relevant problem – Automated refactoring is an active topic bridging ML and software engineering.

Methodologically plausible – Uses established concepts (contrastive learning + RL) in a novel context.

Comprehensive experimental evaluation – Multiple datasets, baselines, and ablations.

Empirical gains are consistent and supported by reasonable qualitative examples.

Clear problem formulation with interpretable reward components and graph structures.

**Weaknesses:**

Writing quality is poor, with several unnatural or incorrect phrases (possibly LLM-edited text).

Limited novelty – the method extends existing ideas (contrastive pre-training, embedding-based rewards) without strong theoretical or algorithmic innovation.

Lack of detailed methodology – unclear how exploration and semantic testing modules are implemented; no algorithm listing or complexity analysis.

Evaluation metrics are not standardized – “Syntactic Improvement” and “Maintainability Gain” are not well defined or justified.

No discussion of failure cases or negative transfer during cross-language testing.

The “Use of LLM” statement at the end suggests possible overreliance on AI-assisted writing without thorough review.

**Questions:**

How is the balance between traditional metrics and embedding dynamics in Eq. (5) determined? Are α, β, γ tuned manually?

Are the embeddings fine-tuned during RL or kept frozen after pre-training?

How are the semantic equivalence tests generated—through symbolic execution or data-driven inference?

What is the computational cost of contrastive pre-training and how scalable is it for multi-language corpora?

Could this framework extend to code optimization or bug repair beyond refactoring?

---

### Meta-Review · Area_Chair_WLyg · 2025-12-09

**Summary:**

This paper proposes a framework called Contrastive Code Graph Embeddings (CCGE) for reinforcement learning–based automated code refactoring. It overcomes the limitations of hand-engineered reward functions by leveraging contrastive pre-training on code graphs and incorporating the learned embeddings into a composite reward function. However, there are major issues with both the method and the presentation. Contrastive pre-training and the composite reward are not described clearly. The authors did not respond during the rebuttal phase. All reviewers recommended against acceptance.

**Reviewer Concerns:**

There are major issues with both the method and the presentation. For example, contrastive pre-training and the composite reward are not described clearly. **The authors did not respond during the rebuttal phase.**

**Reviewer Scores:**

**The authors did not respond, and no discussion during the rebuttal phase.**

---

### Decision · Program_Chairs · 2026-01-26

Reject